# Who Is My Neighbor? Developing a Pedagogical Tool for Teaching Environmental Preaching and Ethics in Online and Hybrid Courses

## Leah D. Schade

Lexington Theological Seminary, Lexington, KY 40503, USA; lschade@lextheo.edu

**Abstract:** As theological education has moved increasingly to online and hybrid settings (both by choice and by pandemic necessity), practical theologians committed to teaching ecological theological education must navigate a paradox. How do we teach about interconnectivity and interdependence between the human and other-than-human inhabitants of a particular place when our classrooms are in disembodied digital spaces? This article examines a case study of a pedagogical tool developed by the author called the "Who Is My Neighbor" Mapping Exercise that enables students to explore and articulate how they conceptualize themselves and their faith communities embedded within their larger ecological contexts. This paper assesses the use of the mapping exercise in four different course contexts: three online and one hybrid online–immersion course. The author provides an overview of each of the four course contexts in which the tool was used, includes descriptions of how students engaged the tool, and assesses its effectiveness. The author uses three types of criteria for assessment of the pedagogical exercise: student feedback, level of competence demonstrated in student assignments (sermons, worship services, teaching events), and personal observations, particularly around the differences between online and onsite contexts. The author suggests that the mapping exercise is a tool that can be used by others teaching practical theology to help students understand their relationships within Creation and their communities, critically engage environmental justice issues, and apply what they learn to their ministry contexts.

**Keywords:** ecological theology; environmental ethics; environmental justice; homiletics; preaching; online teaching; pedagogy; practical theology



## 1. Introduction

As a mainline Protestant homiletician with a passion for environmental issues, I have dedicated a great deal of my teaching, preaching, and writing to equip clergy for "greening" their preaching. A main tenet of my book, *Creation-Crisis Preaching*, is that preachers have a key role to play in helping congregations understand environmental issues as a matter of faith and moral/ethical obligation. However, in my research studying clergy attitudes and opinions around social issues and preaching, I have learned that pastors are sometimes hesitant to address issues such as climate change or species extinction in their sermons. For example, in a 2017 survey of mainline Protestant clergy in the U.S., I learned that environmental issues ranked among the lowest priorities for preachers to address. Only 30% of the 838 respondents said that they had mentioned environmental issues in their sermons in the previous twelve months (Schade 2017). Fortunately, that number rose to 47% when my team and I conducted the second survey wave with 2099 respondents in 2021 (Schade et al. 2021). Nevertheless, this still means that, by their own admission, more than half of the U.S. clergy who responded to this survey are not speaking about climate or environmental issues.

Given the acceleration and intensity of the numerous climate and environmental crises that are devastating communities, one might think that clergy would be willing to bring a prophetic and pastoral word about these issues to their congregations. As Fletcher Harper,

executive director of GreenFaith, has written, "The world needs religious voices that clearly name the causes of this dire crisis, articulate a moral vision, and catalyze courageous action to meet the suffering that lies ahead while bending history's arc towards justice" (Harper 2021). Yet, a number of factors have contributed to the relative silence on environmental issues from the pulpit, including the politicization of topics such as climate change and fossil fuel extraction, as well as the negative pushback that clergy receive for being a "political" preacher when they address such issues.[1]

However, I have also found that many preachers and seminarians simply do not know *how* to bring biblical and theological language to environmental topics and to do so in a way that does not alienate or anger their congregations. Therefore, I have tried to design seminary courses that help students develop an environmentally literate approach to preaching, worship, and faith formation. Yet, as a professor whose courses are either online or hybrid, I recognize that there are challenges to this work when our classrooms are in disembodied digital spaces. So, I have experimented with pedagogical approaches to help students grasp the concepts of interconnectivity and interdependence between the human and other-than-human world.

To that end, this article examines a case study about the use of a pedagogical tool I developed called the "Who Is My Neighbor" Mapping Exercise. This tool is designed to help students explore and articulate how they conceptualize themselves and their congregations embedded within their larger ecological contexts and to share their insights through preaching, leading worship, or teaching in their faith community. I will provide an overview of each of the four course contexts in which I used the tool, include descriptions of how students engaged the tool, and assess its effectiveness. I will use three types of criteria for assessment of the pedagogical exercise: student feedback, level of competence demonstrated in student assignments, and my own observations, particularly around the differences between online and onsite contexts. I suggest that the "Who Is My Neighbor" Mapping Exercise is a tool that can be used by others teaching practical theology courses to help students understand their relationships within Creation and their communities, critically engage environmental justice issues, and apply what they learn to their ministry contexts.

## 2. The Need for Place-Based Ecological Education in Theological Studies

Taylor Tollison has noted that interest in "place-based" education is growing within academia (referencing Ark et al. 2020; Knapp 2014; Morris 2014; Sobel 2004). "In conjunction with this emphasis on place, many institutions are now highlighting not only the ways in which we should learn *from* or *about* a place, but also the ways in which we should learn *for* a place" (Tollison 2021, p. 3). How students and institutions are "embedded" within their communities and ecological contexts has implications not just for the institutions in which they learn, but for their formation as ministers or religious leaders in whatever contexts they serve. How might students "learn *from* the places in which they are embedded in order to learn *for* the places they are embedded?" (Ibid, p. 7).

These are questions I asked when I first began teaching environmental ethics in onsite classrooms. In those settings, I had the opportunity to take students outside to think about their embeddedness and interconnectedness with nature. I used an exercise described by Mark Wallace in his book, *Finding God in the Singing River*, called the Council of All Beings. In this activity, he takes students to a natural area and invites them to seek a connection with one of the lifeforms there. After spending significant time observing, listening, and communing with this lifeform, the group comes back together and shares what they learned by speaking as the lifeform itself. "The purpose of a Council," he explains, "is to foster compassion for all life-forms and heal the ugly splits that separate human beings from the natural world (Wallace 2005, p. 138).

Such imaginative speaking "for" and "with" other-than-human beings can have intriguing implications for preaching, leading worship, and faith formation from an ecological perspective. I have used this exercise for retreats and onsite classes, and it has yielded

insights for participants to integrate into their sermons and their personal relationship with Creation. However, for students in an online classroom, there is no option to be together in one place in nature at the same time. Moreover, unless the Council of All Beings is conducted in the place where a student is engaged in ministry, the exercise is removed from their own context.

So, I wanted to develop a tool to help students root themselves in their own context and create a presentation that they could share both with the class and with their congregation. This would allow students in the digital course to articulate a "theology of place" (Inge 2003) rather than being relegated to a sterile classroom removed from their context. I envisioned an exercise that would enable both instructor and students to traverse digital and local spaces with an eye toward identifying ecojustice issues to address with their congregations. Ideally, such a tool would create a means by which students engaged in distance learning could experience embeddedness and a theology of place within their ecological contexts and then share that through sermons, worship services, and teaching events. I call this tool the "Who Is My Neighbor" Mapping Exercise.

### 3. Developing the "Who Is My Neighbor" Mapping Exercise

I originally developed the "Who Is My Neighbor" Mapping Exercise for preachers to use in their sermon preparation and in ministry with congregants to help them expand the notion of "neighbor" to include other-than-human beings. The goal was to enlarge a preacher's and congregation's circle of care to include other-than-human "persons" who suffer and languish because of environmental injustice. I wanted to give them a tool to survey their landscapes and expand the concept of neighbor to include our Earth-kin so that they could determine what issues need to be addressed in their congregations and communities. I also wanted draw on a biblical story to help frame this work as part of Jesus's mandate to care for those on the margins of our human society, regardless of barriers we might initially put in place.

I chose Luke 10: 25–37 (NRSV) which includes a lawyer's question to Jesus that prompts the parable known as "The Good Samaritan." The lawyer asks what he must do to inherit eternal life. Jesus answers with a question, "What is written in the law? What do you read there?" The lawyer replies by quoting the Torah: Love the Lord your God with all your heart, soul, strength, and mind; and love your neighbor as yourself. But after Jesus affirms his answer, the lawyer follows with a rejoinder: "And who is my neighbor?" Jesus responds with a parable about a Samaritan taking care of a beaten man when two others had ignored him. Jesus then asks, "Which of these three, do you think, was a neighbor to the man who fell into the hands of the robbers?" The lawyer replies, "The one who showed him mercy." The passage ends with Jesus instructing the man to "Go and do likewise."

As I explain in *Creation-Crisis Preaching,* "It is not the qualifications of the one who suffers that determine who should be considered 'neighbor.' It is the one who chooses to care who makes herself or himself a neighbor. In other words, 'neighborness' is not initiated by the one in need of care. It is determined by the one choosing to act in a caring way toward another" (Schade 2015, p. 63). Put another way, the lawyer's question could be restated as, "Who is worthy of my moral consideration? Who qualifies? And who can I justifiably and reasonably exclude?" Jesus turns the question around: "Who will you choose to care about?" In other words, neighborness is not determined by the receiver of compassion, but by the giver. From the Samaritan's perspective, the only qualification for neighborness was the suffering and need of the beaten man. All other lines, walls, hierarchies, and divisions fell away.

Using an ecohermeneutical lens[2] to interpret this passage, I make the case that we must think expansively about the concept of "neighbor" to include our other-than-human kin who, in a sense, lie beaten and bleeding on the side of the road by the "robbers" that have attacked all aspects of Creation. In class discussions, we talk about how we might reconsider this parable about moral obligations within the biotic realm and environmental justice issues. Are animals our neighbors? How about mountains? Ecosystems? The empty

lot down the street filled with trash? The microscopic organisms in the stream killed by fracking waste-water? How far down the food chain and how far afield should we go?

This line of questioning spurs them to think about their own context. Who are the ecological neighbors of our congregation that need our attention and a Samaritan's care? How are people, plants, animals, land, and air suffering from human activity right in our own community? And in what ways might a congregation tend to their wounds with the compassion, attention, and the generosity of the Samaritan?

## 4. Criteria for Determining Eco-Ethical Obligation and Action

Invariably, students raise the thorny question that accompanies all environmental ethics about whether human beings should be prioritized over other lifeforms when it comes to determining moral obligations. As Cynthia Moe Lobeda describes the dilemma:

> Moral consciousness expanded to include the other-than-human is fraught with complexity. To illustrate: . . . On what grounds do specific moral values and obligations apply differently to humans than to other-than-humans? What moral constraints ought be placed on human beings in light of our sameness with and dependence upon otherkind? (Moe-Lobeda 2013, p. 125)

In summarizing John Cobb's critique of deep ecology, Mark Wallace puts it this way: "If all beings—everything from megafauna such as human beings and blue whales to microflora such as mold spores and green algae—are sacred, if everything is equal in value and worth, then on what basis can decisions be made about what should be saved and protected and what can be used and destroyed?" (Wallace 2005, p. 150). In other words, without some sort of hierarchy of values, there would be no way to engage in eco-ethical decision making because value would collapse into sameness.

Wallace's approach to determining how we should make decisions about environmental ethics, which I offer to the students, is to ensure "the health and dynamism of the life cycle rather than protect the interests of added-value beings (such as human beings) whose inner life is more complex than other beings. Thus, green spirituality is able to make highly nuanced and sophisticated practical judgments about use and value, but it does so in biocentric rather than anthropocentric terms" (Wallace 2005, p. 151). He adds that judgments about value "should be based on keeping open the living channels of energy that make life possible . . . [P]ractical decisions about resource allocations and the like should focus on ensuring the dynamism and vitality of the energy cycle, not on the particular needs of individual participants within the cycle, including the needs of individual human participants" (Ibid, pp. 152–53).

Building on Wallace's holistic focus on the life cycle, I suggest to the students that a key for determining ethical action is to apply the criteria of "the least of these", using the term from Matthew 25: 31–46, Jesus' parable of the "sheep and the goats". The "least of these" refers to those most vulnerable who are in need and have little to no voice or agency in their self-preservation. So, for example, on the human level, pregnant women and children are biologically and historically among the most vulnerable of human beings. Their needs would be prioritized when making environmental decisions about pesticides, mercury emissions from coal-fired power plants, and the use of plastics, for example. Applying the criteria to the biotic cycle, the most basic building blocks of life on a cellular and microbiotic level must be considered, and any activity that threatens them should be avoided. For instance, since fracking for methane gas involves the use of chemicals that kill microscopic life and poison the food chain at its most basic level, it should not be allowed.

Thus, rather than determining the rightness of a human activity from the top down (i.e., those in power protecting and promoting their immediate self-interest, gratification, and profit), we would gauge it from the bottom up. In other words, if it is not good for the children, the fish in the sea, or the microbes in the soil, an inverted pyramid of care dictates that it should not be done. Stated in a positive way, the "least of these" are what Jesus has said we are to protect, so whatever promotes their health and well-being fulfills the divine command of caring for them.[3]

**5. "Who Is My Neighbor" Mapping Exercise: Description and Use in the Classroom**

In Chapter 3 of *Creation-Crisis Preaching*, I offer practical suggestions and questions for mapping the ecological, social, cultural, and political location of a particular congregation to help preachers better contextualize their sermons. As I developed the tool, I chose activities that would help participants experience themselves and their congregations embodied within a particular ecological context while also seeing ecological issues related to the larger context of other justice issues. The activities and accompanying questions are designed to help participants think in an interconnected way about the ecological, social, cultural, and political location of a particular congregation in order to better contextualize their ministry.

For the course assignment, I ask students to pick six options from numbers 1–12, and then complete number 13 for their project. I encourage them to do this work in collaboration with a group of parishioners if possible so that the student can incorporate more voices and perspectives. Here are the options (Schade 2015, pp. 73–75):

1.  **Walk**. Walk the grounds around the church building. Consider your surroundings, which include the land, the plants near you, the air you are breathing, and other living creatures perceptible to your senses. Who are your biotic neighbors? Also consider the houses, buildings, businesses, factories, and other human-made "neighbors." Reflect on the interactions that are occurring between you and these multi-faceted surroundings. Are they harmful? Beneficial? Neutral? How do your natural surroundings affect your physical or spiritual existence? Your feelings? Your values?

2.  **Sit**. Choose a location where you can sit and quietly observe and reflect on the interactions that are occurring between you and your multi-faceted surroundings. Allow all of your senses to be engaged as you listen, sniff, and feel the world around you. Where do you perceive harm or pain? What interrelations are beneficial? How do your natural surroundings affect your physical or spiritual existence? Your emotions? Your values?

3.  **Look at a topographical map of where the congregation is located.** Google Maps, Google Earth, or other online mapping services are free and can reveal a bird's eye view of your setting. Notice the local waterways and landscape features (mountains, desert, beach, green spaces, etc.). How are they disrupted, connected to, or otherwise intersecting with human civilization?

4.  **Look at a map that shows the location of major businesses, industries, landfills, waste processing, etc.** Do some research of the basic demographics of the area (census data, income data, sociological data). In what ways do the major "players" interact and have an impact on each other and the biotic community?

5.  **Talk with members of your congregation** to get a sense of "who" (in the expanded ecological sense) their neighbors are, and who has been beaten and lies along side of the road. Who are "the least of these" (Matthew 25: 40) in need of attention and care?

6.  **Talk with other clergy** to learn the history of "neighbor-relations" in the community. What stories do they tell about neighbors helping each other (or not)? Do any of them have shared interest in environmental issues so that you may collaborate on preaching ideas?

7.  **Talk with community members** to hear their stories about environmental issues that are part of the community's history. Were there any grassroots efforts to clean up blighted areas? Protest pollution? Confront toxic dumping? What was successful? What work remains to be done?

8.  **Talk with local health care workers** such as doctors and nurses to find out what the key public health issues affect the community. There are often environmental connections to health concerns. Asthma, obesity, cancer, and depression, for example, are all exacerbated by deleterious environmental conditions such as air pollution, radioactive waste, waste incineration, etc.

9.  **Meet with local chapters of environmental groups** such as Sierra Club, Clean Air Council, Interfaith Power and Light, and grassroots activist groups to find out what

environmental issues they are addressing in your community. Ask how local houses of worship can be helpful to their work.

10. **Talk with local naturalists, master gardeners, those who fish, hunters, farmers, bee-keepers, or others whose work involves the natural elements**. Ask what changes they have observed in animal, plant, insect, fish, or other biotic communities in the last few decades. What would they like to see happen in terms of protecting or sustaining the health of the local ecosystems?

11. **Search for clean-energy businesses in your community** such as wind farms, solar farms, geothermal companies, etc. Inquire as to how they see their work in relation to the community and the planet.

12. **Meet with your elected officials.** Ask them who they consider "the least of these," those most vulnerable among their constituents. What are their main environmental concerns regarding their watersheds, land, forests, and biotic communities within their jurisdictions? What legislation or policies have they supported or opposed for environmental protection?

13. **Create a "map" of your findings.** This can take the form of an actual map with key features noted, a hand-drawn representation, a collage of photos or images, a video, PowerPoint, or some other kind of visual display of the "neighbors" surrounding your congregation. This can be a collaborative project with youth and adults.

Typically, I give students at least a week to complete the steps they choose from numbers 1–12 and then report on their initial findings through an online discussion board. I instruct students to share two to three insights or "a-ha" moments that came from completing the steps and to name the environmental issue they are considering as the topic for their sermon, worship service, or educational event. For the peer responses, I prompt them to ask questions of each other, give feedback as to what they'd like to learn more about, and to seek clarity about the topic their peer plans to address.

For the final project, I have experimented with requiring simple PowerPoint slides, to allowing a variety of formats such as photos of a visual display, videos, or narrated PowerPoints. Students are also required to write a 500–700-word paper in which they reflect on what was most meaningful for them in this exercise as well as their plan for sharing the project with their congregation. As I will describe in the examples that follow, some students presented their "maps" at educational forums, others wrote and preached a sermon, and still others designed a worship service around the theme from their mapping project.

**6. Assessment 1: Lancaster Theological Seminary, Creation-Crisis Preaching Course**

The first time I used the "Who Is My Neighbor" Mapping Exercise in an online course was in the fall of 2018 when I taught Creation-Crisis Preaching for Lancaster Theological Seminary as an adjunct instructor. Seven students registered for the five-week course which was online and asynchronous except for one Zoom meeting at the beginning. After completing readings which introduced them to environmental, theological, and biblical foundations for environmental preaching, the students completed the "Who Is My Neighbor" Mapping Exercise as a step toward their final sermon. The sermon was to be based on Luke 10: 25–37 and needed to connect to a local environmental issue of their choice in their community or region. For some students, this was the first time they were challenged to think about environmental issues at all, much less address one in a sermon. So, I provided a list of twenty-six possible topics ranging from air pollution to waste disposal with the option of identifying a different topic in consultation with me.

While each student completed the mapping assignment, most included just a few slides and only completed four or five of the options in the exercise. One student only submitted a topographical map with some scribbled notes. The sermons were average to above average in content and delivery. I shall highlight two examples.

One student was a White female whose church was situated in a small town in central Pennsylvania. In her mapping exercise, she walked around the church and streets of the town, sat to watch the environmental processes happening around her, talked with

church members, and learned about a new initiative to restore streams and the health of the local water system. These activities were documented in the PowerPoint she created for the mapping assignment. In her sermon on the parable of the Good Samaritan, she began by talking about the idea of neighbors in a general way while referencing the South African concept of *ubuntu*, meaning "I am, because you are." Then she moved to expand the concept of neighbor to include God's Creation, noting that human beings can only exist because the rest of nature exists. This brought her to the "neighbor" suffering in their own community—the streams that run through their town and surrounding countryside that have been deteriorating due to nitrogen and chemicals from agricultural and yard runoff as well as silt accumulation. She made the case that the example of the Samaritan's extravagant service toward his neighbor urges us to be extravagant in the care we give our water system. She encouraged that care to be collective, systemic, and engaging local policy. However, she did not go as far as encouraging them to attend local meetings and express their support for water restoration projects or to engage in the projects themselves. So, in my feedback to her, I suggested that she offer those next steps to them in future conversations and sermons.

In contrast, another student, who was a Black female, served a church that was located in an urban area of Baltimore, Md. Through her mapping exercise as documented by her PowerPoint, she walked around the neighborhood and took pictures of deteriorating buildings, looked at city maps, talked with local residents and clergy, and educated herself about the effects of lead-based paint on the health of children. Building on the work her church was already doing to address the need for affordable, safe housing for local families, she drew three key points from the Lukan passage on the Good Samaritan: (1) stop (do not be a passerby), (2) do what you can, and (3) invite others to help. In my feedback, I recommended that she be even more specific and invitational by encouraging the congregation to become more involved in local efforts to clean up the neighborhood and support healthy homes for their friends and neighbors. I suggested that she identify projects that are happening in the community and invite the congregation to join in what she called "God's Samaritan work" in a tangible, concrete way.

Overall, while I received positive feedback from students about the mapping exercise, I felt that I had not clearly articulated my expectations for the final projects. This resulted in less than stellar submissions. So, I knew I needed to be more specific with the instructions when I used the tool in the future.

## 7. Assessment 2: Lexington Theological Seminary, Witness and Testimony in Appalachia Course

The second time I used the mapping exercise was for the course, Witness and Testimony in Appalachia, in the summer of 2019 at Lexington Theological Seminary where I serve as Associate Professor of Preaching and Worship.[4] I received a grant from the Appalachian Ministries Educational Resource Center (AMERC) to provide scholarships and funding for the course which covered transportation, lodging, and meals for twelve students.[5] The course was a hybrid of online and onsite learning elements. Before arriving for the immersion trip, students engaged online readings, videos, podcasts, and completed a short research paper about the coal region of eastern Kentucky. The four-day trip then provided the opportunity for students to learn first-hand about the complex ways in which geography, class, culture, race/ethnicity, and sexual orientation impact the conjoining challenges of environmental, socioeconomic, and public health issues in the region.

The students were assigned to create the "Who Is My Neighbor" Map to reflect the people we met (such as coal miners, high school students, clergy and their congregations, and local community organizers), and places and environmental "neighbors" we visited, including an historic coal mine, an old growth forest protected by Kentucky Natural Lands Trust, and mountains and waterways destroyed by mining. This time, I had the students consider guiding questions when thinking about their mapping assignments: what is it like to be you in this place? What would it be like if you could envision a positive future?

Whose stories do we tell? Who gets to tell these stories and how will those stories be told? These questions enabled students to engage in deep, pastoral listening to a variety of voices (including other-than-human neighbors in God's Creation), and to discern how they can preach the gospel in the midst of contentious community issues.

Unlike the first time I used the exercise where most students submitted a few PowerPoint slides, this time I received many different types of maps, including collages of labeled photographs, images overlayed across a map of the region, a radial chart, and a poster with the image of a web in the background to indicate the web of life. I intended the mapping exercise to help them organize their ideas for preaching a sermon when they returned to their home congregations. The sermon was to highlight what they learned in the Appalachian immersion experience and to demonstrate that they had thought critically about how one tells the story of a people and culture. Two of the more intriguing sermons were preached by students whose countries of origin were outside the U.S.

One was a male student who was a native of Ghana and served an African-native congregation in Louisville, Ky. He drew parallels between the way the environment is exploited for coal in Appalachia and how the rivers of his own country are polluted because of mining. He used the story of the Woman at the Well in John 4:4-14 to encourage his congregation to advocate for "living water".

Another student was a female native of Haiti serving a Haitian congregation in New York City. She noted that both the residents of Appalachia and the residents of Haiti are like the man in the parable of the good Samaritan in Luke—beaten, robbed, and left to die on the side of the mountain. She also connected the "neighbor" thread from the drug addicts in their neighborhood, to the mother who does not have enough money to feed her children, to the trees cut down in their homeland, to the sea crying out from being suffocated with trash. She encouraged her congregation to be a "Good Samaritan church" by teaching the youth to love God's Creation and by joining with their interfaith neighbors to clean up abandoned lots in their community.

I was much more pleased with the mapping assignment submissions the second time around. Not only were students creative and detailed in their maps, they also expressed appreciation for having a tangible reminder of the trip. Many of the submissions were so impressive that I decided to use them as examples to show students in the future when using this tool.

## 8. Assessment 3: Hartford Seminary, Environmental Ethics Course

In the third iteration of the "Who Is My Neighbor" Mapping Exercise, the course was not about preaching and worship but environmental ethics. I taught the online, asynchronous, semester-long course for Hartford Seminary in the spring of 2020. Unlike the first two Christian seminaries where I used the exercise, this was an interfaith seminary which meant that I had sixteen students from Christian, Muslim, Unitarian, non-denominational, and secular contexts. This made for a rich learning environment for the students and for myself.

Instead of using the mapping exercise as a lead-in for preaching or designing worship, I made it a stand-alone assignment and had the students present their mapping exercise to their faith community as an educational event. Because this course coincided with the emergence of the coronavirus pandemic that was causing houses of worship to close, I allowed the students to create narrated PowerPoints which could be shown during an online education event or accessed by congregation members asynchronously.

One student was a Muslim male in Houston, Tx, who began his presentation by comparing the parable of the good Samaritan with a teaching from An-Nisaa 4:36 about caring for "neighbors who are near, neighbors who are strangers, the companion by your side, the wayfarer (ye meet)." His PowerPoint then showed the wilderness of concrete and pavement around their masjid from the street view and from a Google Earth view. He zeroed in on one underdeveloped area of green space and encouraged his fellow

congregants to join an effort to revitalize a local plan for parks, recreation, and open space in Harris County.

Another student was a White female faculty member at a co-educational boarding and day school for high school students in Connecticut. She used the mapping assignment to highlight not just the community and environmental neighbors, but to incorporate the voices of Indigenous Peoples. As she wrote in her reflection paper, "This project has inspired me to bring a proposal for land acknowledgement, education, and professional development to school leadership. I've been working with another colleague, and we've actually been making some great progress! We secured the funding to bring in a speaker from a Native community here in Connecticut." Her PowerPoint consisted of twenty-six well-designed and informative slides that incorporated nearly all of the twelve options from the mapping exercise and concluded with concrete "next steps", including a presentation the following month called "Dialogues on Difference" for the community.

Other student projects included preserving green spaces and partnering with local environmental groups in the church's community, water conservation for congregation members as well as the church's building and grounds, protecting a community against wildfires, and becoming a "zero waste" congregation. One student even created a live-action video about the lifecycle of a single-use plastic bottle. The video showed the interrelatedness of "neighbors" ranging from the oil that is used to produce the bottle, to the recycling facility, to the places where "recycled plastic" actually ends up—oceans, underdeveloped nations, and in the stomachs of fish and sea birds.[6]

This time around, the students were able to see models of the mapping exercise that I had collected from the previous course, and I believe this made a difference in the quality of their assignments. In all, the mapping exercise for this course provided a way for students to develop critical skills for theological analysis of and creative engagement with environmental challenges that their faith communities could address. The exercise also empowered them to provide education and, for some students, leadership on the issue they chose to address.

## 9. Assessment 4: Lexington Theological Seminary, Creation-Crisis Preaching Course

The fourth time I utilized the mapping exercise was in May 2021 at Lexington Theological Seminary. Twenty-two students registered for the four-week online course, sixteen of whom were part of a continuing education certificate program, "The Church and Creation", and were not required to submit assignments. This course was about incorporating environmental issues into both preaching and worship, so in addition to the sermon assignment, students were also required to design a Creation-centered worship service complete with prayers, hymns, readings, and a communion liturgy.[7] The "Who Is My Neighbor" Mapping Exercise, then, was intended to situate their worship design and sermon within the political, cultural, and biotic setting of the church.[8]

One White male student served a congregation in the Cumberland Plateau of Tennessee and focused on the "neighbors" of the East Fork of the Obed River and Dale Hollow Lake. He made it a point to speak with beekeepers, farmers, hunters, those who fish, winemakers, and others who work in natural elements. He referenced two biblical texts in his sermon. He used Isaiah 24: 4–5 ("The earth dries up and withers") to describe the pollution and neglect that the river suffers by the hands of humans. However, he also lifted up a vision of hope in Matt. 6: 25–33 ("Look at the birds of the air"). "Jesus' words serve as an example of looking to nature, so that we might learn from those other-than-human life forms; learn how-to live-in harmony with our biotic neighbors," he said in the sermon. He encouraged the congregation to care for "the least of these" through supporting local environmental and recreational groups, scout troops, and community organizations that are already working to revitalize the river and lake. He framed this as "environmental healing ministry" at the conclusion of the sermon. The student also designed a worship service with a theme of "taking care of God's *oikos* (household)", with hymns and prayers for Creation in general and rivers and water in particular.

Two time zones away, another student created her mapping assignment for her desert-dwelling congregation in Arizona. This White female student created a video highlighting the native plants, trees and birds, the local senior living facility, an elementary school, a food bank, and a community farm demonstrating desert food production and ecological restoration. She even showed pictures of the very edge of the church's property where a homeless encampment had left remnants of their stay that day. She stressed that all of these were the church's neighbors and that she intended to focus on the "circular, cyclical, self-balancing design of Creation" as symbolized by the church's large outdoor labyrinth. She preached her sermon outside standing in front of the labyrinth and drew parallels between the "deserted place" where Jesus fed the multitudes in Mark 6: 30–44, and the "deserted place" there in the Sonoran Desert, which had thrived ecologically before European colonization. She encouraged the congregation to listen to the teaching of Creation and of Jesus Christ to see the "hidden abundance" all around them. She concluded, "Jesus bids us to offer what we have, then sit down with all of Creation and be fed." She, too, designed a worship service complete with Creation-centered hymns, prayers, and other liturgical elements.

In this course, I noticed a marked improvement in the mapping exercises and sermons compared to those submitted the first time I taught the course in 2018. For instance, most of the PowerPoints in 2018 had 5–10 slides. For this course in 2021, however, the number of slides ranged from 10–24, included much more text and graphics, and showed evidence of completing six or more of the steps in the exercise. I think one of the reasons for the improved quality of the assignments was due to the fact that I provided examples of mapping exercises completed by students in the Appalachia immersion course and the Hartford course so that students could get ideas for their own projects. I found that the students engaged with much more depth, detail, and diligence in the project than the first time I assigned the exercise three years prior.

## 10. Possibilities for Future Research and Use in Teaching Contexts

While students have expressed unanimous appreciation for the "Who Is My Neighbor" Mapping Exercise, I do not know how their congregations responded or what long-term effects the projects had on the students' ministries. In the future, one way to gauge the effectiveness of this exercise would be to conduct a before-and-after survey of the congregation. Prior to the project, the congregation would receive a questionnaire asking about their attitudes toward religion and ecology as well as their knowledge about the topic the student would be addressing in their mapping exercise. Then, after the student completes the project, a second survey would be distributed asking the same questions as well as additional questions about how they felt moved to respond or put their faith into action in light of the student's sermon, worship service, or educational presentation. In this way, the comparative data would provide feedback not only about the student's individual project, but about the overall effectiveness of using the mapping exercise.

There is also the potential to develop a version of the exercise that guides students and members of a congregation to envision ways they can address environmental issues more fully as a community of faith. Working with a team in their congregation, the student could choose just one or two of the steps in the exercise as learning activities to be done with their congregants that would then inform the student's map-making. I would also welcome the opportunity to see others adapt the "Who Is My Neighbor" Mapping Exercise for Roman Catholic and Orthodox contexts within Christianity, as well as in other faith traditions such as Judaism or Islam.

## 11. Conclusions

In whatever ways students in theological education choose to address environmental issues in their ministries, I remind them that this work does make a difference. Research has shown that homilies and sermons can enhance the effectiveness of the message to embrace green Christianity. For example, the 2014 PRRI/AAR survey on religion, values, and climate

change found that "Americans who say their clergy leader speaks at least occasionally about climate change are more likely to be climate change Believers than Americans who tend not to hear about climate change in church (49% and 36%, respectively)" (Jones et al. 2014, p. 4). In other words, people in the pews are listening to what their ministers have to say about climate and other environmental issues. Simply making the case that Creation-care is part of our responsibility as Christians is a vital message that clergy and faith leaders can and should convey.

As I reflect on the use of the "Who Is My Neighbor" Mapping Exercise for these four courses over the span of three years, I have come to see that the tool was a source of spiritual formation for many students. I encouraged them to prayerfully consider how the process of reading scripture, interpreting faith through an ecological lens, and attending to the suffering of their human neighbors as well as Earth kin could inform and shape their own faith and relationship with the Divine. In turn, they critically engaged matters of faith and ecological justice in local and global contexts.

The mapping exercise is also a useful reference point as I invite students to consider what concrete actions they might take on environmental issues beyond the conclusion of the course. Especially as they move beyond an individualistic framework (What should *I* do?) toward a more congregationally and community-based framework (What should *we* do?), the mapping exercise is a way for students to engage people in their houses of worship and local community on these questions.

As the examples above demonstrate, the mapping exercise also encouraged students to learn about the ways in which race, socioeconomics, culture, religion, and local context relate to and with environmental issues. Therefore, I suggest that this exercise could be used in any number of teaching contexts, courses, and disciplines to help inculcate a "theology of place" and emphasize the importance of learning "*from* the places in which they are embedded in order to learn *for* the places they are embedded" (Tollison 2021). For example, this mapping exercise could be used in courses on food and faith, climate ethics, God and nature, environmental law and policy, biodiversity and nature's rights, ecological racism, and climate migration and the church, to name just a few. The "Who Is My Neighbor" Mapping Exercise is a project that can enable students to develop an expansive and holistic understanding of environmental issues while making the case that Jesus' teaching about showing mercy extends to our biotic neighbors as well. "Go and do likewise."

**Funding:** This research was partially supported by a grant from the Appalachian Ministries Educational Resource Center which provided funding for students in the course, Witness and Testimony in Appalachia, Lexington Theological Seminary, June 2019.

**Institutional Review Board Statement:** Not applicable.

**Informed Consent Statement:** Not applicable.

**Conflicts of Interest:** The author declares no conflict of interest.

## Notes

[1] In both the 2017 and 2021 survey, half of respondents said they received negative pushback in the form of angry emails, letters, or direct confrontation when preaching about social issues.

[2] Norman C. Habel suggests six guiding ecojustice principles: the principle of intrinsic worth, the principle of interconnectedness, the principle of voice, the principle of purpose, the principle of mutual custodianship, and the principle of resistance (Habel 2000, p. 2).

[3] According to Paul W. Taylor, "William Frakena delineated eight types of ethical theories which could generate moral rules and/or judgments concerning how rational agents should act with regard to the natural environment. The eight types are differentiated by their conceptions of moral subjects or patients. Each has its own view of the class of entities with respect to which moral agents can have duties and responsibilities. The eight types may be briefly delineated as follows: 1. Only what benefits or harms the agent himself is morally relevant to how anything else in existence should be treated. (Egoism.) 2. Only humans (or those humans who are also persons) are proper moral patients. How we ought to act with respect to the environment is determined ultimately by the effects of our actions on humans or on persons. 3. All conscious (or sentient) beings are proper moral patients. Conduct with regard to the environment is right if it alleviates the suffering or increases the pleasure of beings that can suffer or

experience pleasure. 4. All living beings, conscious or not, are proper moral patients. Our moral concern should extend beyond humans to all animals and plants. 5. Everything in existence (other than God), whether taken distributively or collectively, is to be considered as that toward which we may have duties and responsibilities. 6. God is the only ultimate moral subject as far as human action is concerned. We owe duties only to God, and we should treat the natural world in such a way as to fulfill our duties to God. 7. Combinations of any two or more of the above. 8. Nature itself is a moral patient. We should either follow the ways of nature or let the ways of nature take their course without our intervention." (Taylor 1981). My assessment is that Wallace prioritizes #5. The "Who Is My Neighbor" Mapping Exercise utilizes a combination of #4, #5, and #6.

[4] Appalachia is a region of the United States spanning thirteen states across the Appalachian Mountains from southern New York to Mississippi. While blessed with beautiful mountains, valleys, forests, rivers, lakes, and abundant natural resources, some areas have suffered from environmental devastation, poverty, and public health issues (such as lung disease and addictions). In addition, harmful classist and cultural stereotypes, as well as tensions around race, ethnicity, and sexual orientation further complicate attitudes within and about the people of Appalachia.

[5] The mission of AMERC is to promote contextual, cross-cultural education for theological students, faculties, and other Christian leaders. Working primarily through an ecumenical consortium of theological schools, regional colleges or universities, oversight agencies of the church, and supporting organizations, AMERC supports experiential learning about the theological, spiritual, social, economic, and environmental aspects of Appalachian culture, especially for ministry in rural and small-town settings.

[6] The video can be accessed on Youtube: https://www.youtube.com/watch?v=NT4IJBiO7ws (accessed on 28 March 2022).

[7] In addition to *Creation-Crisis Preaching*, two other required texts were: *Liturgies from Below: Praying with People at the Ends of the World* by Cláudio Carvalhaes (Nashville, T, Abingdon Press, 2020), and *A Watered Garden: Christian Worship and Earth's Ecology* by Benjamin Stewart (Minneapolis, MN: Augsburg Fortress, 2011). A recommended text was *The Season of Creation: A Preaching Commentary* edited by Norman C. Habel„ David M. Rhoads, and H. Paul Santmire (Minneapolis, MN: Fortress Press, 2011).

[8] I also provided several website resources to the students to give them ideas for designing their worship service and preaching their sermon, including: Creation Justice Ministries resources for Earth Day. http://www.creationjustice.org/urgency.html (accessed on 28 March 2022); Let All Creation Praise, http://www.letallcreationpraise.org/ (accessed on 28 March 2022); EcoAmerica, https://ecoamerica.org/ (accessed on 28 March 2022); Blessed Tomorrow, https://blessedtomorrow.org/ (accessed on 28 March 2022; Climate Health, https://ecoamerica.org/health/ (accessed on 28 March 2022); Lutherans Restoring Creation, http://www.lutheransrestoringcreation.org/ (accessed on 28 March 2022); and Emerging Earth Community, http://www.emergingearthcommunity.org (accessed on 28 March 2022).

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
