# Peer review of "Who Is My Neighbor? Developing a Pedagogical Tool for Teaching Environmental Preaching and Ethics in Online and Hybrid Courses"

_religions, doi:10.3390/rel13040322_

Round 1

Reviewer 1 Report

I think it is a great exercise and well-written. Excellent.

Author Response

Thank you!

Reviewer 2 Report

The author writes clearly and present his/her argument very well. The article contributes to contemporary discussions about care for creation and theological education for environmental sustainability. Since care for creation is the most important socio-moral issue of our times, this is an important article to publish.

I have two substantive comments concerning possible revisions to the article.

  1. I suggest that the author develop more fully his/her argument in lines 142-155 concerning our moral obligations to the neighbor. The essential question to consider is: Do we as human beings have greater moral obligations to persons than we do to other forms of life, including and perhaps especially other forms of animal life? As a more embodied instance of this question, I might ask: Do I have a greater moral obligation to the human neighbor who lives next door than I do to the trees in my front yard and the fish, ducks, and beavers who live in the pond across the road? Essentially, for the author’s argument to be fully developed, she/he needs to say something about how she/he thinks we should see our human neighbors in relation to our non-human neighbors.

The issue I am raising is, of course, complex and multi-faceted, and the author could not discuss it fully. Yet, I suggest that the author indicate an awareness of the issue and offer a foundational response. As a resource for addressing the issue the author might review the articles in the Journal of Environmental Ethics. One early yet foundational and still relevant discussion of the issue is found in William Frankena’s 1981 article in the Monist, and republished elsewhere since then, titled “Ethics and the Environment.”

  1. Has the author considered how she/might revise the “Who is our Neighbor?” Mapping Exercise or tailor it for specific audiences? If so, might the author include some of her/his reflections about this issue in section five on future possibilities?

The fact that most participants in learning situations in all of the instances the author used the exercise did not complete all 13 steps suggests to me that the exercise is too long for many if not most learners to process fully. This could lead the author to ask questions such as: What steps in the learning activity are most important for preachers? …for religious educators and teachers? …for parish pastoral ministers? …for members of a congregation? In the instances in which the exercise was used what steps were most engaging for students and why?

Might the author consider developing a version of the exercise that is aimed specifically at helping preachers incorporate concern for care of creation into their preaching? Might the author develop a version of the exercise that guides members of a congregation to envision ways they can as a community of faith address environmental issues more fully? In a course, could steps 9 through 12 be separate learning activities done in groups, partly in-class and partly out of class, that would then inform participants’ map making.

I suggest the author address the following minor issues.

  • In line 14, “in which used the tool was used” should be “in which the tool was used”
  • I suggest the author check the quote that is found in both line 80 and line 483. Is the word placed correct, that is, is that how the quote reads in the text of origin? If so, I suggest that the author include (sic) after the word.
  • In line 185, “How are they are” should be “How are they”
  • In live 304, “a provided” should be “provided”
  • In line 431, a common is needed after “Desert”
  • It is not fully clear to me what the author means in line 489 by “weave the intersectionality of environmental issues.” Could the author explain what she/he means a bit more fully? It is trendy at present to include discussions of intersectionality in academic discourse. Perhaps the author might consider quoting a source that has informed his/her understanding of intersectionality? In any case, in her/his few brief references to intersectionality, the author needs to make sure that what he/she is saying about intersectionality is stated clearly.

I offer the following for further reflection:

Given the authors’ interest in theological education, he/she might be interested in the Religious Education Association (see their website) and the journal Religious Education. There have been a few notable essays on environmental education in the journal from the early 90s to the present.

Some of the language the author uses suggests that he/she has a background in Protestant theology. To expand his/her horizon the author might consider incorporating into his future work, analyses of the environmental crised from Catholic and Orthodox perspectives. The US Catholic bishops have two statements on the environment. The first universal statement by the Roman Catholic Church on environmental responsibility was Pope John Paul II’s 1980 World Day of Peace message on peace with creation. Pope Benedict XVI discussed our obligation as persons to care for the environment in several of his works and statements, and Orthodox Patriarch Bartholomew has been an outspoken advocate for addressing the environmental crisis. More recently, in Laudato si’ Pope Francis issued a call for all people to engage in efforts to care for the natural environment.

From a pedagogical perspective, I ask: Rather than ending with map mapping could the author in future scholarly work and teaching end the learning exercise with an activity that invites students to decision and action, that is, that asks students what they could and will do to contribute to efforts to address the contemporary environmental crisis?

Author Response

Thank you for these helpful suggestions, many of which I have incorporated into the next revision.  

Regarding this comment: "The fact that most participants in learning situations in all of the instances the author used the exercise did not complete all 13 steps suggests to me that the exercise is too long for many if not most learners to process fully. "

To clarify, in line 166, I state:  "For the course assignment, I ask students to pick six options from numbers 1 – 12, and then complete number 13 for their project."  

In other words, the students are not expected to do all 13.

Thank you for such a robust engagement with this paper!